# INTERCHANGEABLE TOKEN EMBEDDINGS FOR EXTENDABLE VOCABULARY AND ALPHA-EQUIVALENCE

## ABSTRACT

We propose a novel approach for learning interchangeable tokens in language models to obtain an extendable vocabulary that can generalize to new tokens. Our method addresses alpha-equivalence, the principle that renaming bound variables preserves semantics. This property arises in many formal languages such as temporal logics, where all proposition symbols represent the same concept but remain distinct. To handle such tokens, we develop a dual-part embedding approach. The first part is shared across all interchangeable tokens, enforcing that they represent the same core concept. The second part is randomly generated for each token, enabling distinguishability. As a baseline, we consider a simpler approach that uses alpha-renaming for data augmentation. We also present alpha-covariance, a metric for measuring robustness against alpha-conversions. When evaluated in a Transformer encoder-decoder model for solving linear temporal logic formulae and copying with extendable vocabulary, our method demonstrates promising generalization capabilities as well as a favorable inductive bias for alpha-equivalence.

## 1 INTRODUCTION

Following the deep learning revolution that affected numerous application areas (Dargan et al., 2020), recent literature shows that deep learning based approaches also perform well in neurosymbolic reasoning tasks, such as theorem proving (Han et al., 2021) and mathematical reasoning (Rabe et al., 2020). The formal reasoning capabilities of these models were once doubted, but Liu et al. (2023) demonstrated the ability of Transformer models (Vaswani et al., 2017) to learn shortcuts to automata. Of particular interest is the generalization ability of such models to unseen, out-of-distribution data (Sanh et al., 2021), enhancing their appeal for logical reasoning (Abbe et al., 2023).

Another application area is linear-time temporal logic (LTL), which is heavily utilized by the formal verification community (Clarke et al., 2018; Baier & Katoen, 2008) for reasoning about how logical propositions change over time (Pnueli, 1977). Through the use of temporal operators, LTL formulae can specify, for example, that a proposition $p$ must hold at all time steps ($\mathbf{G}p$), or at least one time step ($\mathbf{F}p$). LTL formulae operate on traces, which describe how the propositions change over time.

Solving a given LTL formula involves finding a satisfying trace, and it proved essential for generating examples for system specifications in the literature. This field was dominated by the methods that use classical algorithms, such as spot (Duret-Lutz et al., 2022) and aalta (Li et al., 2014). However, following the success of Transformer models on end-to-end symbolic integration (Lample & Charton, 2019), Hahn et al. (2021) attacked the LTL solving problem using the same approach. Their capability to generalize to longer formulae is especially noteworthy, and it was made possible thanks to tree-positional encoding (Shiv & Quirk, 2019).

However, generalization to longer formula lengths is not the only concern. In particular, each LTL formula features a set of atomic propositions (henceforth APs), and it's desirable for the model to generalize to more APs. But the architecture of the model does not even accept new APs that are not seen during training, despite the fact that all APs represent *semantically equivalent* concepts while being *distinguishable* from each other. This situation arises in many other application areas, such as mathematical expressions and lambda calculus (alp, 1984), where renaming the bound variables does not change the meaning. This phenomenon is described as *alpha-equivalence*. *Alpha-conversion* (or *alpha-renaming*) refers to the process of creating alpha-equivalent input-output pairs.

In this paper, we propose a novel approach for representing interchangeable tokens in neural network models. To summarize, our method constructs some part of the token embeddings on-the-fly instead of learning all of them during training. The token embeddings for interchangeable tokens consist of two parts: a learnable part and a randomized part. The learnable part is shared across all interchangeable tokens, and the model must depend on the randomized part to differentiate these tokens. We use the weight tying technique (Press & Wolf, 2016) to share the same token embeddings with the final projection matrix, which calculates the logits (i.e., next-token probabilities before softmax).

We use our embedding method in a Transformer encoder-decoder model and evaluate it on two tasks: copying with an extendable vocabulary and solving LTL formulae. For the second task, we use datasets generated using `spot` (Duret-Lutz et al., 2022), a library for LTL manipulation and model checking using conventional algorithms. As a baseline, we consider a simpler approach that uses alpha-renaming for data augmentation during training to expose the model to a larger vocabulary, which is also new in the literature to the best of our knowledge. Overall, our method demonstrates generalization capabilities to larger vocabulary sizes, and also combines well with positional encodings that exhibit length generalization. We also experiment with dataset perturbation to show that our method introduces a helpful inductive bias for alpha-equivalence. Finally, we present *alpha-covariance*, a metric for measuring robustness against alpha-conversions.

## 2 RELATED WORK

**Language modeling and formal reasoning**   The transformer architecture (Vaswani et al., 2017), now ubiquitous in modern deep learning, was initially proposed as a generative model to translate between natural languages autoregressively. This led to many successful attempts to frame formal reasoning tasks as language modeling problems, such as symbolic integration (Lample & Charton, 2019), symbolic regression (Kamienny et al., 2022; Vastl et al., 2022), LTL solving (Hahn et al., 2021), and many more. Further developments shifted the field towards large language models (LLMs), e.g., by prompting a model pre-trained on a gigantic scale (Frieder et al., 2023), by enhancing the prompt with retrieved references for proof generation (Welleck et al., 2022; Yang et al., 2023), by training an LLM on a specialized dataset for mathematics (Azerbayev et al., 2023). However, the reasoning abilities of LLMs were questioned by (Tang et al., 2023), who showed LLMs struggle with symbolic reasoning when semantics are decoupled, and by others (Wu et al., 2023).

**Extensible vocabulary**   Efforts to create an extensible vocabulary for neural networks are scarce in the broader machine learning community, let alone the formal reasoning literature. Morazzoni et al. (2023) exploited dictionary definitions to create extensible word embeddings. Wei et al. (2016) proposed a vocabulary-extensible sign language recognition framework by using a component based approach, where each sign gesture is recognized based on common components such as hand shape, orientation, axis, rotation, and trajectory. These studies depend on either external information (dictionary definitions) or properties specific to an application area (components of hand gesture); they do not attempt to design an extensible vocabulary for interchangeable tokens, which has been neglected by the literature alongside the concept of alpha-equivalence.

## 3 PROBLEM DEFINITION

In a language modeling problem, the goal is to predict the next token in the output sequence given the input and the past output. (See Appendix A.1 for more details on language models.) Let $\mathbb{V}$ denote the set of all unique tokens, i.e., the vocabulary of a language modeling problem. We assume that $\mathbb{V}_i$ is the set of interchangeable tokens and $\mathbb{V}_n = \mathbb{V} \backslash \mathbb{V}_i$ is the set of non-interchangeable tokens. The core idea behind *alpha-equivalence* is that renaming interchangeable tokens between each other in both input and output preserves meaning. Let $f \colon \mathbb{V} \to \mathbb{V}$ be a bijection such that $f(x) = x$ for all $x \in \mathbb{V}_n$, i.e., $f$ arbitrarily renames the interchangeable tokens between each other in one-to-one correspondence and preserves the rest of the tokens. We apply $f$ to each token in a given pair of input $\boldsymbol{a} \in \mathbb{V}^*$ and output $\boldsymbol{b} \in \mathbb{V}^*$ strings, obtaining $\boldsymbol{a}' = (f(a_1), f(a_2), \ldots)$ and $\boldsymbol{b}' = (f(b_1), f(b_2), \ldots)$. We call this operation *alpha-conversion* or *alpha-renaming*. The set of interchangeable tokens $\mathbb{V}_i$ must be defined such that $\boldsymbol{a}'$ and $\boldsymbol{b}'$ form a valid input-output pair semantically equivalent to $(\boldsymbol{a}, \boldsymbol{b})$ for all possible $f$.

Our task is to design an embedding method that—alongside being resilient to alpha-renaming by construction—can support a new vocabulary $\mathbb{V}' = \mathbb{V}'_i \cup \mathbb{V}_n$ where $\mathbb{V}_i \subset \mathbb{V}'_i$ after training on $\mathbb{V}$. In other words, the model should be able to operate on a larger vocabulary than the one seen during training, as long as the newly introduced tokens belong to the class of interchangeable tokens. Although we don't impose any restrictions about the size of $\mathbb{V}'$ in this problem definition, the maximum size of $\mathbb{V}'$ in practice may change as a function of the number of embedding dimensions. Thus, while setting the hyperparameters, the expected size of $\mathbb{V}'$ must be considered.

**Example** In the LTL solving problem (Appendix A.2), the set of non-interchangeable tokens $\mathbb{V}_n$ includes the operators, constants, delimiter tokens ("`;`", "`{`", "`}`"), and any special tokens such as the end token. The set of interchangeable tokens equals to the set of atomic propositions (APs): $\mathbb{V}_i = P$. Assuming $P = \{a, b\}$, the formula-trace pair ("`&aXb`", "`a;b;{1}`") is alpha-equivalent to ("`&bXa`", "`b;a;{1}`"). Further, assume that the augmented set of interchangeable tokens is $\mathbb{V}'_i = P' = \{a, b, c, d\}$. Now, the aforementioned pair can also be equivalently represented as ("`&cXd`", "`c;d;{1}`"). The augmented vocabulary allows the expression of formula-trace pairs that feature up to 4 APs instead of 2. For example, ("`&&abX&cd`", "`&ab;&cd;{1}`") cannot be expressed using $P = \{a, b\}$. Our goal is to create a model that can handle such inputs despite being trained on the limited vocabulary $\mathbb{V} = \mathbb{V}_n \cup P$.

# 4 PROPOSED METHOD

To address the problem of learning semantically equivalent but distinguishable (alpha-equivalent) tokens, our method employs two ideas: sharing some part of the embeddings between such tokens to convey their semantic equivalence; and assigning a unique randomly-generated vector to the rest of the embedding for each interchangeable token, allowing the model to distinguish between them. The number of shared and randomly-generated dimensions are denoted by $d_\alpha$ and $d_\beta$ respectively. The sum of these two yields the total number of embedding dimensions in the model, denoted by $d_{\text{model}} = d_\alpha + d_\beta$. For non-interchangeable tokens, $d_\alpha$ dimensions contain separate learnable parameters and $d_\beta$ dimensions are set to 0. The structure of the embedding matrix is visualized in Figure 1.

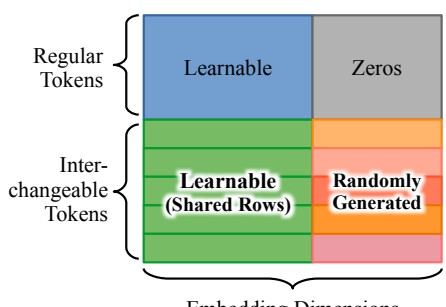

Figure 1: Visual structure of the embedding matrix in the proposed method.

## 4.1 EMBEDDING MATRIX

**Construction of the embedding matrix** For a vocabulary with $n$ non-interchangeable tokens and $m$ interchangeable tokens, $\boldsymbol{L} \in \mathbb{R}^{n \times d_\alpha}$ represents the matrix of learnable embeddings for non-interchangeable tokens, $\boldsymbol{\alpha} \in \mathbb{R}^{1 \times d_\alpha}$ the shared learnable embedding for interchangeable tokens, and $\boldsymbol{\beta}_i \in \mathbb{R}^{1 \times d_\beta}$ the randomly-generated embedding for the $i$th interchangeable token where $1 \leq i \leq m$. Note that $\boldsymbol{\alpha}$ and $\boldsymbol{\beta}_i$ are row vectors. A zero matrix of size $i \times j$ is represented by $\boldsymbol{0}^{i,j}$. In addition, we define two row-based L2 normalization functions $f_{bn}(\boldsymbol{X})$ and $f_{fn}(\boldsymbol{X})$ that divide each row $\boldsymbol{X}_{i,:}$ by its L2 norm $||\boldsymbol{X}_{i,:}||$. These two functions are identical but can be disabled independently from each other, hence the separation. Finally, the overall structure of the embedding matrix $\boldsymbol{U}$ is shown in Equation 1. In this construction, the interchangeable tokens are assumed to come after the non-interchangeable tokens. Note that it's also possible to implement multiple sets of different interchangeable tokens via a trivial extension.

$$
\boldsymbol{U} = f_{fn}(\begin{bmatrix} f_{bn}(\boldsymbol{L}) & \boldsymbol{0}^{n,d_\beta} \\ f_{bn}(\boldsymbol{\alpha}) & f_{bn}(\boldsymbol{\beta}_1) \\ f_{bn}(\boldsymbol{\alpha}) & f_{bn}(\boldsymbol{\beta}_2) \\ & \vdots \\ f_{bn}(\boldsymbol{\alpha}) & f_{bn}(\boldsymbol{\beta}_m) \end{bmatrix}) \tag{1}
$$

During training, the embedding matrix must be reconstructed in each forward pass with resampled random vectors $\boldsymbol{\beta}_1$ to $\boldsymbol{\beta}_m$. Resampling $\boldsymbol{\beta}_i$ for $1 \leq i \leq m$ during training prevents the model from adapting to the idiosyncracies of a particular random generation and forces it to distinguish between interchangeable tokens regardless of the contents of $\boldsymbol{\beta}_i$. During inference, it's created once at the start and remains the same since the autoregressive generation involves multiple forward passes on the same input.

**Normalization**    There are several concerns that warrant the heavy use of normalization while constructing $\boldsymbol{U}$, as seen in Equation 1. Firstly, $d_\alpha$ dimensions and $d_\beta$ dimensions should not overwhelm each other in terms of magnitude. Normalizing $\boldsymbol{\alpha}$ and $\boldsymbol{\beta}_i$ separately addresses this issue. The magnitude of the concatenated embedding is another concern, which is handled by the final normalization. The normalization of $\boldsymbol{L}$ is redundant (since the final normalization does the same operation after the concatenation with zeros) but kept in Equation 1 for readability.

## 4.2    RANDOM EMBEDDING GENERATION

This section will explain how the distinguishing part of the interchangeable token embeddings, $\boldsymbol{\beta}_i, 1 \leq i \leq m$, are created. To this end, we developed 3 methods to generate random vectors. Table 1 provides a summary at a glance. The first method simply samples the standard normal distribution for each dimension. The second one uses the neighboring grid points around the origin, which correspond to the 8 directions in 2D. For each interchangeable token, a unique vector in this set is sampled. The last method is similar, but its set consists of the vertices of a hypercube centered around the origin, i.e., diagonal direction vectors.

Table 1: Comparison of random vector generation methods.

| Method | Normal Distribution | Neighboring Points | Hypercube Vertices |
|---|---|---|---|
| **Formula** | $\mathbf{a}_i \sim \mathcal{N}(0, 1)$ | $\mathbf{a}_i \in \{-1, 0, 1\}$ $\lvert\mathbf{a}_i\rvert \neq 0$ | $\mathbf{a}_i \in \{-1, 1\}$ |
| **Size** for $n$-dims | Continuous | $3^n - 1$ | $2^n$ |
| **Sample Visualization** |  |  |  |

In the normal distribution method, we don't have any additional constraints to ensure distinguishability between vectors. However, in other two methods, we need make sure that each interchangeable token gets assigned to a unique vector since the sampling set is finite. To achieve this quickly and space-efficiently, we define a mapping from integers to possible vectors. The unique vectors are generated by sampling $m$ unique random integers (which can be calculated efficiently using the reservoir sampling technique), and then using the defined mapping to convert these integers to the vectors. This strategy avoids materializing the whole set of possible vectors. In the hypercube vertices method, we map the binary digits of an integer in $[0, 2^{d_\beta})$ to $\{-1, 1\}$. Although "Neighboring Points" is simply the ternary version of the same idea, avoiding the zero vector requires special care. The zero vector maps to the integer $i_z = (3^{d_\beta} - 1)/2$. Therefore, we define our domain as the integers in $[0, 3^{d_\beta} - 1)$ and add 1 to the integer $i$ before converting it if $i \geq i_z$.

Integer mapping approach for generating unique vectors works well for up to 32 dimensions, after which the limits of integer representation become an issue for reservoir sampling. Therefore, in such cases, we simply disable the uniqueness check because the size of the sampling set grows exponentially, rendering the probability of drawing the same sample negligible.

## 4.3    PROJECTION

**Weight tying**    In a traditional language modeling setting, since both the embedding and projection matrices are entirely composed of learnable parameters, it's not necessary to share them, even

though there are many advantages of weight tying (Press & Wolf, 2016). However, we construct the embedding matrix manually in our method, which makes weight tying a requirement. Furthermore, since we perform our experiments on an encoder-decoder architecture in this paper, we utilize a three-way weight tying approach, whereby the embedding matrices of encoder and decoder are tied in addition to the final projection matrix. Three-way weight tying is particularly appropriate for our problem domain since many tokens are shared between the LTL formulae and traces.

**Feature normalization**   Given the output of the last layer before the final projection $v$ (henceforth called feature vector), instead of directly applying the final projection as in $Uv$, we apply L2 normalization to the feature vector $v$ before passing it through the final projection: $Uf_{fn}(v)$. This matrix multiplication constitutes taking a dot product with each row. Since $a \cdot b = |a||b|cos(\theta)$ where $\theta$ is the angle between $a$ and $b$, normalizing both the embeddings and the feature vector leaves only the cosine term to determine the logits. This forces the model to distinguish between tokens based solely on the directions, which may improve the gradient flow.

**Cosine loss**   If we normalize both the embeddings and the feature vector, the only thing that determines each logit is the cosine of the angle between the feature vector and the embedding. Applying the softmax loss to such logits is known as cosine loss in the literature. Although cosine-based loss functions were successful in face recognition (Ranjan et al., 2017; Wang et al., 2017), it proved sensitive to hyperparameter settings in these losses. To avoid this problem, we use AdaCos loss function (Zhang et al., 2019) that scales the logits adaptively throughout training.

To adapt the AdaCos loss function to our use case, we make the following modifications: Since the language modeling problem involves a sequence length dimension in addition to the batch dimension, we combine these two dimensions while ignoring the padding tokens, effectively treating both dimensions as batch dimensions. However, since this change greatly increases the number of batch dimensions, it can lead to numerical issues, even with the log-sum-exp trick. Therefore, we clip the scale value calculated by AdaCos to a maximum of 100 to avoid numerical issues.

## 5    EXPERIMENTS

**Experimental setup**   We use a transformer encoder-decoder architecture in all experiments. We always use the same embedding size in both encoder and decoder due to weight tying. We use the RoPE (Su et al., 2023) as the positional encoding method in both encoder and decoder unless otherwise noted. The hyperparameter settings are given in Table 4 in Section A.3.

**Baselines**   We train three types of baseline models with traditional embeddings: the first one on the original dataset, the second one on a dataset with the same parameters but using a larger vocabulary size, and the third one on the original dataset but using a data augmentation strategy. Specifically, for the third baseline, the number of interchangeable token embeddings matches that of the test set, and we apply random alpha-renaming at each forward pass during training. This ensures that the model is exposed to all tokens in the test set, but the number of unique interchangeable tokens the model sees in each sample remains limited as in the training set. Note that this is an internal baseline that doesn't exist in the literature to the best of our knowledge.

### 5.1    COPYING WITH EXTENDABLE VOCABULARY

We introduce a new toy problem designed to evaluate the vocabulary generalization capabilities of our embedding method. We create various training datasets that contain 10 million random strings with a limited vocabulary size. A string is given as input, and the model is expected to produce the input string exactly via autoregressive generation. This embodies a helpful toy problem for our method because all tokens are interchangeable, barring the special tokens (start/end). In these experiments, we expect the model to generalize to larger vocabulary sizes unseen during training. We generate the predictions using greedy sampling in this subsection.

**Evaluation method**   We use the edit distance between the prediction and the ground truth as our evaluation metric. To generate the evaluation datasets (validation and test splits), we create 100 samples for each possible combination of unique character count and string length, starting from

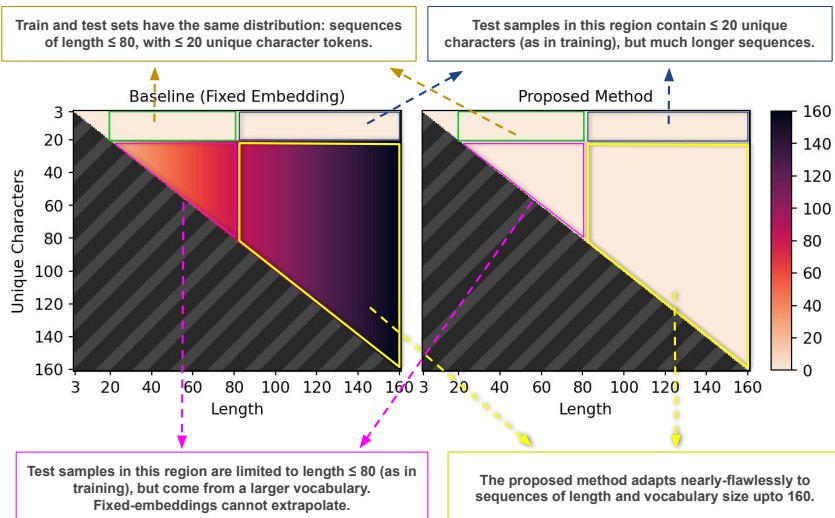

Figure 2: Two annotated heatmaps visualizing the test-set edit distance between prediction and ground truth in copying task with extendable vocabulary. Both heatmaps share the same y-axis. The green box represents the number of unique characters (y-axis) and the maximum length (x-axis) in the training dataset. The lower triangular part of each heatmap, shown in gray hatch pattern, represents the impossible combinations of length and unique character count. Each point represents the average error over test samples with a particular sequence length and unique number of character tokens. The baseline approach (on the left), using ubiquitously utilized fixed (learned) token embeddings, cannot extrapolate to vocabulary expansions. The proposed method (on the right) enables generalization to larger vocabulary sizes at longer sequence lengths, compared to what is observed during training.

a minimum of 3. Consequently, the total evaluation dataset is arranged in a matrix in which the rows represent unique character count in the string and the columns represent the string length. This matrix is upper triangular since the unique character count cannot exceed the string length. For random embeddings, we repeat the evaluation 10 times and report the average. To evaluate up to the string length of 30 in this setup, $10 \times 100 \times 406 = 406000$ predictions are required, where $406$ is the number of upper triangular elements in a $28 \times 28$ matrix. To minimize the impact of random factors, we train each model three times and report the results only for the best.

### 5.1.1 GENERALIZATION TO LARGER VOCABULARIES

We create a dataset consisting of 10 million strings whose lengths vary between 3 and 30 with at most 5 unique characters. We evaluate the models on strings up to length 30 with at most 30 unique characters. Out of 27 models we trained with dual-part embeddings, 20 of them achieve an average edit distance of 0.0, i.e., no error. The worst model's average edit distance is 1.0. For comparison, an output sequence of length 30 can have a maximum edit distance of 30.

### 5.1.2 GENERALIZATION TO LARGER VOCABULARIES AND LENGTHS

We create a dataset consisting of 10 million strings whose lengths vary between 5 and 10 with at most 5 unique characters. We evaluate on the same validation set as before, expecting the model to generalize to both longer lengths and larger vocabulary sizes. In Appendix A.3.1, we perform a hyperparameter search over random embedding methods, $d_\beta$ values, and whether $f_{bn}$, $f_{fn}$, AdaCos are enabled.

We determine the best model for the proposed method and the baseline on the validation set, evaluate them on the test set and visualize the results in Figure 3. Since the baseline model cannot process larger vocabularies, we assume that the prediction is empty if the unique character count exceeds the training set's vocabulary, hence the edit distance equals length in that area. Our best model trained on limited length uses Hypercube Vertices with $d_\beta$ set to 6 and $f_{fn}$ + AdaCos enabled. It achieves a

mean edit distance of 0.38 on the test set. The first baseline's mean edit distance is 0.51 (calculated up to 5 unique characters, only for this model). The second and third baselines' mean edit distances are 4.93 and 1.85 respectively. However, the significance of this difference is highly questionable, as these models exhibit high variance across different training runs.

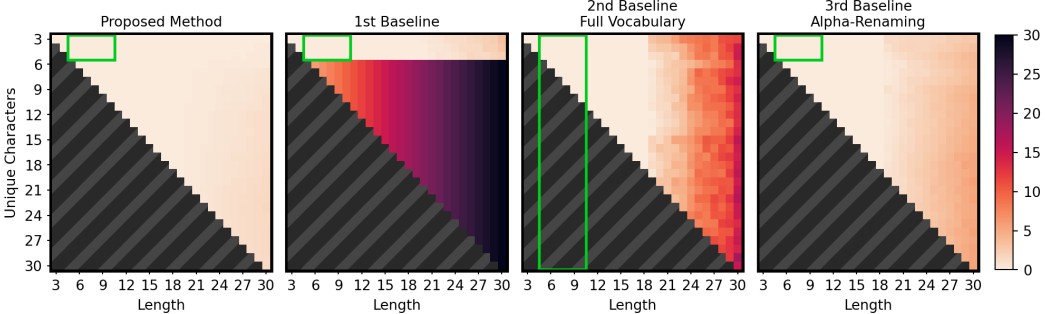

Figure 3: Edit distance heatmaps on test set. The first and second heatmaps are the proposed and baseline (first type) models respectively, trained on strings up to length 10 and a vocabulary size 5. The third heatmap is the second baseline, which uses a new training dataset with a larger vocabulary. The last heatmap is the third baseline that uses the same dataset as the proposed method but incorporates alpha-renaming in training. The difference between the last two baselines is that the alpha-renaming baseline is not exposed to more than 5 unique characters per sample. The lower triangular part of each heatmap (gray hatch pattern) represents the impossible combinations of length and unique character count. The green box represents the number of unique characters (y-axis) and the maximum length (x-axis) in the training dataset. Note that all heatmaps share the same y-axis.

### 5.1.3 SENSITIVITY TO RANDOMNESS IN EMBEDDINGS

We analyze the impact of the randomization that the proposed method performs on embeddings. The minimum, mean, and maximum edit distance (on test set) obtained by ten different embedding randomizations of the second model in Figure 3 are 0.25, 0.38, 0.55 respectively, with a sample standard deviation of 0.09. The pooled standard deviation of the edit distance across all 277 models evaluated on the validation set is 1.73. However, our best models are more resilient against randomness: this value is 0.74 for top 10% models.

To reduce the computational cost of evaluation in the next experiments, we generate 10 random embeddings, sort them by their cross entropy loss on the evaluated dataset, and use the median one. We find that this serves as a decent proxy for the average performance. Across the validation set evaluations of all 277 models, the percent difference in edit distance between this median method and the real mean is 1.4% on average (meaning that the result from the median method is worse), and 9.1% if we consider the absolute differences.

### 5.1.4 SCALING UP

We increase the length of the strings from 5-10 to 20-80, and vocabulary size from 5 to 20. We create the evaluation sets by generating 20 samples for each combination of unique character count and string length. The mean edit distance of our best model is 0.0. The heatmap is given in Figure 2. All baselines also attain perfect performance in this task on the vocabulary sizes they support. Therefore, only the first type of baseline is shown in Figure 2.

### 5.2 LTL SOLVING

In this section, we train models on the LTLRandom35 dataset from DeepLTL (Hahn et al., 2021) and other synthetic datasets created with the same method. To evaluate the correctness of the generated formulae, we utilize `spot` framework version `2.11.6` (Duret-Lutz et al., 2022). We use tree-positional encoding (Shiv & Quirk, 2019) in the encoder and RoPE (Su et al., 2023) in the decoder. We generate predictions using beam search with a beam size of 3 in this section.

**Baselines** We trained all of the baseline models from scratch. For the first type of baseline, we aimed to reproduce the results from Hahn et al. (2021). Hence, we used the best hyperparameters they reported (Appendix A.3). Unlike Hahn et al. (2021), we experimented with RoPE (in the decoder) and AdaCos, but did not observe a noteworthy improvement on the validation set. [1] After determining the best baseline model on the validation set, we evaluated it on the test split of LTL-Random35 and obtained a correct rate of 98.2% against the 98.5% reported by Hahn et al. (2021).

### 5.2.1 DATASET PERTURBATIONS

To demonstrate that our method creates a helpful inductive bias, we created a perturbed version of the LTLRandom35 dataset by renaming the APs such that the order of the first AP appearances in the trace is always the same. As the empirical evidence in Table 2 confirms, both our method and the alpha-renaming baseline are naturally immune to these alterations. We train these methods only on the perturbed dataset since training them again on the normal dataset amounts to training with different random samples.

Table 2: Evaluation of the baselines and our method trained on different versions of LTLRandom35. The alpha-renaming baseline was trained using 5 AP embeddings since vocabulary generalization is not valuated here. First two columns denote the training dataset and the model. Next two columns indicate the ratio of the correct predictions and exact matches on 99,989 test set samples as evaluated by `spot`. Last three columns display mean alpha-covariance values for varying atomic proposition (AP) counts, evaluated on all alpha-equivalent variants of 1000 test samples.

| Training Dataset | Model | Evaluation | | Alpha-Covariance | | |
|---|---|---|---|---|---|---|
| | | Correct | Exact | 3 AP | 4 AP | 5 AP |
| Normal | Baseline | 98.23% | 83.23% | 96.87% | 95.86% | 91.80% |
| Perturbed | Baseline | 34.13% | 12.12% | 64.93% | 57.99% | 40.91% |
| Perturbed | Alpha-Renaming | **97.96%** | **77.66%** | **99.55%** | **99.49%** | **98.86%** |
| Perturbed | Proposed | 95.94% | 76.45% | 97.66% | 97.76% | 98.29% |

While the original model performs significantly worse under perturbation, both alpha-renaming and proposed models match the baseline performance in correctness ratio despite perturbation. This observation suggests that these modifications introduce a robust inductive bias that makes the models resistant to perturbations in the data. A minor decrease in the ratio of exact matches is noted, but this may signify less overfitting and a better bias-variance tradeoff in the larger context. Appendix A.4 continues this experiment with limited amount of training samples instead of perturbations.

### 5.2.2 ALPHA-COVARIANCE

Given a vocabulary of $n$ AP tokens and an LTL formula-trace pair containing $k$ APs, it's possible to write $^nP_k = n!/(n-k)!$ alpha-equivalent pairs. Since these are semantically equivalent, we expect the model's predictions to be the same after undoing the alpha-conversions for all of them. As there is no metric to quantify this in the literature to the best of our knowledge, we develop a new metric.

Let $(\boldsymbol{x}, \boldsymbol{y})$ be an input-output pair for the model, and let $\mathbb{P} = \{(\boldsymbol{x}^1, \boldsymbol{y}^1), \dots, (\boldsymbol{x}^n, \boldsymbol{y}^n)\}$ be $n$ input-output pairs alpha-equivalent to $(\boldsymbol{x}, \boldsymbol{y})$. We define $\alpha_i$ as the alpha-conversion function for the $i$th input-output pair such that $\alpha_i(\boldsymbol{x}) = \boldsymbol{x}^i$ and $\alpha_i(\boldsymbol{y}) = \boldsymbol{y}^i$. To compute the alpha-covariance of a model with respect to $\mathbb{P}$, we generate predictions for each input in $\mathbb{P}$, obtaining the prediction $\hat{\boldsymbol{y}}^i$ for each $x^i$. We define a set that contains the predictions with alpha-conversion undone: $\mathbb{U} = \{\alpha_i^{-1}(\hat{\boldsymbol{y}}^i) \mid 1 \leq i \leq n\}$. Note that if we defined this set for the ground truth outputs in $\mathbb{P}$, we would get $\{\boldsymbol{y}\}$ since $\alpha_i^{-1}(\boldsymbol{y}^i) = \boldsymbol{y}$ holds for each $\boldsymbol{y}^i$ by definition. The model's sensitivity to alpha-conversions could be quantified by simply $|\mathbb{U}|$, but this value may be hard to interpret since it depends on $|\mathbb{P}|$. To normalize this value intuitively, we define the alpha-covariance of a model with respect to $\mathbb{P}$ as in Equation 2.

$$1 - \frac{|\mathbb{U}| - 1}{|\mathbb{P}| - 1} \tag{2}$$

---

[1] Using RoPE in the decoder increased the ratio of correct predictions from 97.8% to 98.0% on the validation set. Introducing AdaCos in addition to RoPE increased this value to 98.2%.

Intuitively, when alpha-covariance is 1, the model is unaffected by all alpha-conversions in $\mathbb{P}$. An alpha-covariance of 0 indicates that $|\mathbb{U}| = |\mathbb{P}|$, i.e., the model's prediction for each alpha-equivalent pair is unique after undoing the alpha-conversion. This is unwanted because alpha-conversions should not change the semantic meaning. Thanks to the embedding randomization in our method, an alpha-conversion does not necessarily change the embeddings, and conversely, there are multiple ways to embed the same input due to randomness.

For the proposed method, we generate the random embeddings once at the start of an evaluation run using the heuristic explained in Section 5.1.3. Thus, alpha-conversions in this experiment are equivalent to shuffling the random embeddings in our method. As a result, the alpha-covariance measures our model's robustness against differences in random embeddings.

We report the results in Table 2, which demonstrates that our method has a positive impact on the alpha-covariance, especially in limited data settings. Since the LTLRandom35 dataset was created synthetically, it doesn't have any noteworthy biases and even the baseline enjoys a high alpha-covariance thanks to this. However, when the dataset is perturbed by introducing a bias to the order of APs, the baseline struggles heavily with alpha-covariance, whereas our method does not.

### 5.2.3 GENERALIZATION

The test dataset for this experiment contains at most 100 formula-trace pairs for each combination of AP count and formula length, whose maximum is 50 instead of 35. We report the results for our model (using Hypercube Vertices, $d_\beta = 5$) and the three baselines in Figure 4. The first baseline uses the same training dataset, whereas the second baseline uses a new LTL dataset with 10 APs, which we create using the same method as LTLRandom35. For the third baseline, we train a fixed embedding model with 10 APs using the same 5 AP dataset but we shuffle the AP embeddings in each forward pass during training.

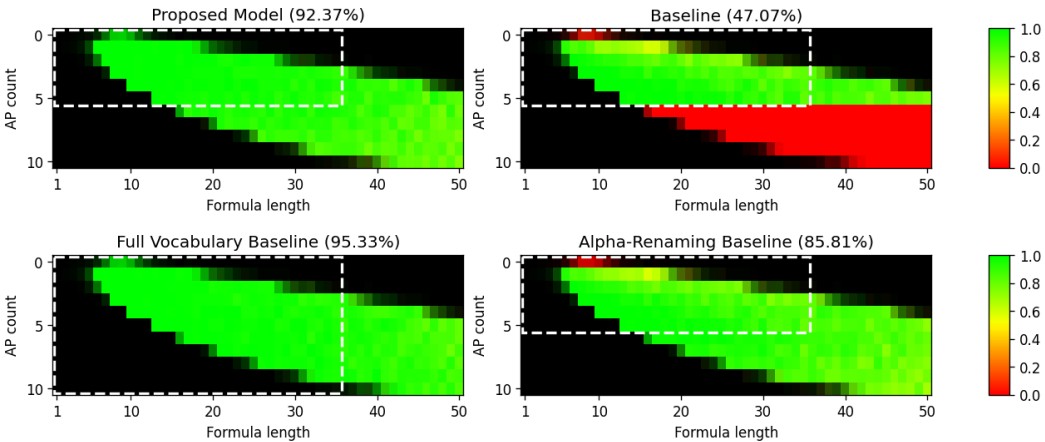

Figure 4: Heatmap visualizing the ratio of correct predictions on a special test set. The brightness of the color depends on the sample size, with full brightness representing 100 samples. The dashed white box represents the boundaries of the training dataset.

**Discussion** Despite seeing only 5 APs during training, our method performs only slightly worse than the full vocabulary baseline, which represents what a transformer-based model can do with 10 APs. Our method outperforms both the vanilla and the alpha-renaming baselines by a considerable margin, which is significant since the latter is the only other model that can generalize to more APs. Based on this, we hypothesize that the proposed stochastic AP embeddings provide a more explicit enforcement towards learning embedding-covariant transformations in the model, as opposed to training with alpha-renaming, where the learned embeddings may still carry unwanted token-specific biases. Furthermore, unlike the baseline models, our model does not have to learn the concept of AP from scratch for each AP token thanks to the shared embedding part. This could explain why our method shone against the alpha-renaming baseline in the LTL task where the interchangeable tokens are more complex than the copying task.

**Motivation for generalization**   The generalization to larger AP counts is important especially when considering the exponential growth of the dataset generation time. In Figure 5, we visualize the growth pattern of the trace checking duration based on increasing formula length and AP count. The times are relative to the fastest trace checking time. The exact times will vary depending on the machine. In our experiments, generating 100000 samples of exact formula length 50 with at most 10 APs took 2 hours and 21 minutes on a system with 56 threads.

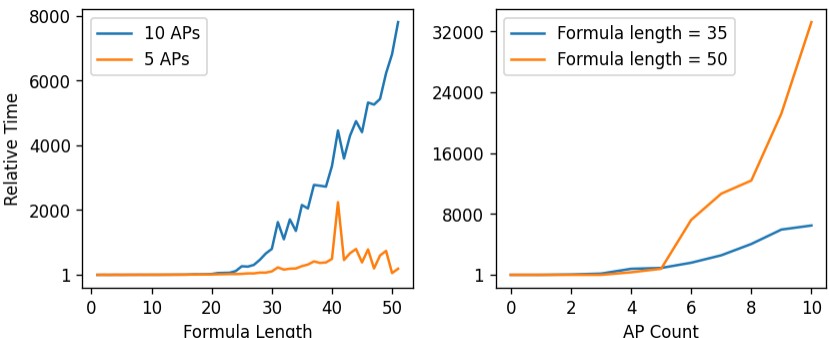

Figure 5: Scaling behavior of the trace generation using `spot`.

**Alpha-covariance**   We evaluate the alpha-covariance performance of these models in Table 3. Note that since 10 APs lead to a lot more naming permutations than 5 APs, the alpha-covariance values are remarkably smaller compared to Table 2. Unlike the results from Table 2, however, our method outperforms the alpha-renaming approach here. This shows that our method excels in out-of-distribution settings, but trades off some in-distribution performance. Although the full vocabulary baseline performs very similarly to our method, it's important to note that this region is in-distribution for that model. Overall, these results align with Figure 4.

Table 3: Mean alpha-covariance values for varying AP counts, evaluated on 1000 test samples, each with 120 random alpha-equivalent variants. The best value for each AP count is highlighted in bold.

| Model | **Alpha-Covariance** | | | | | | | |
|---|---|---|---|---|---|---|---|---|
|  | **3 AP** | **4 AP** | **5 AP** | **6 AP** | **7 AP** | **8 AP** | **9 AP** | **10 AP** |
| Full Vocabulary | 54.09% | 45.51% | 45.23% | **42.07**% | 33.54% | 34.47% | 32.36% | **28.42**% |
| Alpha-Renaming | 50.64% | 43.00% | 40.95% | 37.49% | 30.80% | 30.30% | 28.76% | 25.57% |
| Proposed | **54.30**% | **46.05**% | **45.64**% | 41.88% | **33.89**% | **35.29**% | **33.18**% | 28.34% |

## 6   CONCLUSION

The primary difference between machine learning and numerical optimization is the intention to generalize to out-of-distribution samples, for which the network architecture and its inductive biases play a vital role. In this work, we addressed the challenge of generalizing to larger vocabulary sizes unseen during training and creating an inductive bias for alpha-equivalence. We also contributed the alpha-covariance metric for measuring the model's consistency against alpha-equivalent inputs. These contributions embody a foundation for learning extensible vocabularies for interchangeable tokens, which is especially useful for formal reasoning tasks in which alpha-equivalence naturally arises. Although our dual-part embedding method demonstrates generalization capabilities, its performance in the LTL solving task decreases slightly in in-distribution data (Table 2). The future work can tackle this issue, which may eventually lead to Pareto improvements in bias-variance tradeoff. Moreover, applying our approach to problems in which the interchangeable tokens have meaningful names (e.g., human-written variable names) represents an intriguing area for future research. Finally, new randomization and/or normalization methods for our embeddings can be explored.

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

# A    Appendix

## A.1    Preliminary: Language models

The autoregressive language modeling or sequence modeling in a broader sense—whose goal is to predict the next token given the past tokens—was revolutionized by the transformer architecture (Vaswani et al., 2017), replacing the step-by-step processing of recurrent neural networks (RNNs) with a parallelizable attention mechanism. At its core lies the attention mechanism, which computes three vectors—query, key, and value—from input embeddings. This mechanism allows the model to weigh the importance of different tokens, enabling it to capture long-range dependencies efficiently. In self-attention, these vectors come from the same sequence, while in cross-attention, key and value vectors come from a different sequence, as in encoder-decoder setups. The transformer consists of an encoder with self-attention and feed-forward layers, and a decoder that adds cross-attention to incorporate the encoder's output. Since attention lacks an inherent sense of token order, positional encodings are added to input embeddings to provide sequence structure. During training, attention masking ensures causality in predictions, preventing future tokens from being considered when predicting the next one.

## A.2    Temporal logic overview

Linear Temporal Logic (LTL) extends propositional logic by introducing the ability to reason about the evolution of propositions over time (Pnueli, 1977). The syntax of LTL, defined over a finite set of atomic propositions $P$, is as follows: The syntax of LTL, defined over a finite set of atomic propositions $P$, is given in Equation 3, where $\mathbf{T}$ represents *True*, $p \in P$ an atomic proposition, $\neg$ the negation operator, $\wedge$ the conjunction operator, $\mathbf{X}$ and $\mathbf{U}$ the temporal operators *next* and *until* respectively.

$$\phi := \mathbf{T} \mid p \mid \neg\phi \mid \phi_1 \wedge \phi_2 \mid \mathbf{X}\phi \mid \phi_1 \mathbf{U} \phi_2 \tag{3}$$

Specifically:

- $\mathbf{X}\phi$ holds at time $t$ if and only if $\phi$ holds at the next time step, i.e., at time $t+1$.

- $\phi_1 \mathbf{U} \phi_2$ means that $\phi_2$ must hold at some future time $t_2$, and $\phi_1$ holds at every time step $t$ from the current time $t_1$ up to but not necessarily including $t_2$.

For instance, the formula $\mathbf{X}\mathbf{X}a$ specifies that $a$ must hold at the third time step. Similarly, the formula $\mathbf{T}\mathbf{U}a$ requires that $a$ holds at some point in the future. Finally, as a more complex example,

the formula $\mathbf{X}b \wedge a\mathbf{U}c$ asserts that $b$ holds at the second time step, $c$ holds at some future time, and $a$ holds at all preceding time steps.

An LTL formula is evaluated over a *trace*, which represents a sequence of truth values for atomic propositions over time. In this work, as in DeepLTL (Hahn et al., 2021), we consider *symbolic* traces of *infinite* length. These traces are expressed in what is known as a *lasso* form, denoted $uv^\omega$, where $u$ is a finite prefix, and $v$ is a finite sequence that repeats indefinitely.

A symbolic trace represents all traces that satisfy the propositional formulae at the respective time steps. For example, the symbolic trace $a, a \wedge \neg b, (c)^\omega$ describes all traces in which $a$ holds at the first two time steps, $b$ does not hold at the second time step, and $c$ holds at every step from the third onward. This symbolic trace satisfies the formulae $\mathbf{T}\mathbf{U}c$ and $\mathbf{X}\neg b \wedge a\mathbf{U}c$, but it violates the formula $\mathbf{X}\mathbf{X}b$ since $b$ is not guaranteed to hold at the third time step. Symbolic traces, such as this one, can be underspecified, meaning that certain propositions (e.g., $a$ and $b$) may take arbitrary values at some time steps.

The LTL solving problem involves identifying a symbolic trace in lasso form $uv^\omega$ that satisfies a given input formula $\phi$. We approach this as an autoregressive language modeling task: given an LTL formula and a partially generated symbolic trace, the model predicts the probabilities for the next token in the trace.

For compatibility with the dataset from DeepLTL (Hahn et al., 2021), both our traces and formulae are represented in Polish (prefix) notation, where operators precede their operands. For instance, $a \wedge b$ is written as `\&ab`, which avoids the need for parentheses to resolve ambiguities.

As described earlier, we assume that traces are infinite and represented in lasso form $uv^\omega$. Alongside atomic propositions, constants (`True:1` and `False:0`), and logical operators, we use special symbols in the notation: "`;`" is a position delimiter, and "`{`" and "`}`" enclose the repeating period $v$. For example, the string "`a;\&ab;{b}`" represents the symbolic trace $a, a \wedge b, (b)^\omega$.

### A.3 HYPERPARAMETERS

The constant hyperparameter choices for all experiments are given in Table 4. These hyperparameters are kept constant within an experiment. The hyperparameters for the LTL task is taken from DeepLTL (Hahn et al., 2021).

Table 4: Hyperparameter choices.

| Experiment | Embedding | Layers | Heads | FC size | Batch Size | Train Steps |
|---|---|---|---|---|---|---|
| Copy (Sections 5.1.1 and 5.1.2) | 64 | 2 | 4 | 64 | 512 | 20K |
| Copy (Section 5.1.4) | 128 | 6 | 8 | 128 | 512 | 20K |
| LTL (Section 5.2) | 128 | 8 | 8 | 1024 | 768 | 52K |

### A.3.1 HYPERPARAMETER SEARCH

On the smaller copying task, we train multiple models that use different random embedding methods (Section 4.2) with different $d_\beta$ values. While altering $d_\beta$, we keep the total number of embedding dimensions $d_\alpha + d_\beta$ constant. We train each model at least 3 times with different seeds and report the results for the best one in Tables 5 (proposed method) and 6 (baselines).

Table 5: Mean edit distance for various models using proposed method. The numbers in the header row represents $d_\beta$ for each random embedding method. In the first column, enabled normalization features are listed. AC refers to AdaCos, which can only be enabled when $f_{fn}$ is used.

| Enabled Features | Normal Distribution | | | | | Neighboring Points | | | | | Hypercube Vertices | | | | |
|---|---|---|---|---|---|---|---|---|---|---|---|---|---|---|---|
| | 2 | 4 | 8 | 16 | 32 | 4 | 6 | 8 | 16 | 32 | 5 | 6 | 8 | 16 | 32 |
| $f_{bn} + f_{fn} + AC$ | 13.6 | 5.4 | 4.6 | 8.1 | 8.1 | 1.9 | 13.0 | 2.2 | 1.0 | 2.1 | 2.8 | 0.4 | 7.5 | 8.4 | 3.9 |
| $f_{fn} + AC$ | 7.6 | 13.1 | 4.6 | 2.2 | 5.2 | 8.7 | 11.5 | 2.8 | 2.9 | 2.2 | 0.5 | 3.7 | 3.2 | 4.2 | 4.1 |
| $f_{bn} + f_{fn}$ | 13.7 | 10.6 | 8.3 | 3.8 | 11.8 | 11.9 | 5.7 | 3.7 | 7.4 | 8.3 | 2.2 | 13.1 | 21.5 | 19.4 | 20.9 |
| $f_{fn}$ | 15.4 | 10.6 | 8.2 | 3.7 | 10.1 | 8.1 | 12.3 | 6.4 | 13.4 | 9.9 | 2.5 | 1.7 | 12.5 | 2.1 | 12.8 |
| $f_{bn}$ | 10.6 | 16.6 | 11.8 | 6.9 | 8.2 | 5.8 | 3.0 | 0.6 | 7.8 | 14.3 | 12.8 | 13.8 | 19.4 | 22.9 | 11.6 |
| - | 16.5 | 11.6 | 12.6 | 12.5 | 9.0 | 12.5 | 3.7 | 9.5 | 5.9 | 13.5 | 12.7 | 9.6 | 8.6 | 15.9 | 16.6 |

Table 6: Mean edit distance for various baseline models. In the first column, enabled normalization features are listed. AC refers to AdaCos, which can only be enabled when $f_{fn}$ is used. Note that $f_{bn}$ is not applicable for baseline models. The results for the first type of baseline are omitted since it cannot generalize to larger vocabularies. The second baseline was trained on a dataset with a vocabulary size of 30. The third baseline uses the same limited vocabulary dataset like the proposed method, but uses alpha-renaming as data augmentation.

| Enabled Features | Baseline 2nd Type | Baseline 3rd Type |
|---|---|---|
| $f_{fn}$ + AC | 6.1 | 1.9 |
| $f_{fn}$ | 4.9 | 11.3 |
| - | 5.5 | 12.9 |

The results in Tables 5 and 6 exhibit high variance with no clear patterns that indicate which methods are better. Therefore, we perform an analysis based on correlation coefficients between these hyper-parameters and the edit distance using the results from all 277 models we've trained (not including the baseline models). For this analysis, we assume that the value of Boolean properties (such as $f_{bn}$, $f_{fn}$ and AdaCos) are 0 or 1. The correlation coefficients are as follows:

| N.D. | N.P. | H.V. | $d_\beta$ | $f_{bn}$ | $f_{fn}$ | AdaCos |
|---|---|---|---|---|---|---|
| 0.02 | -0.14 | 0.11 | 0.01 | 0.10 | -0.29 | -0.41 |

First three columns are the random embedding methods as listed in Table 1, the fourth column is $d_\beta$, and the last three columns represent whether the given feature is enabled. Accordingly, the best random embedding method is "Neighboring Points" since it's the only one that correlates negatively with edit distance. The correlation observed for $d_\beta$ is negligible. Introducing $f_{bn}$ increases the edit distance, but the statistical significance is not ideal (p-value 0.04). Both $f_{fn}$ and AdaCos loss have a positive and statistically significant impact on edit distance, with p-values smaller than $10^{-6}$.

## A.4 LTL EXPERIMENT WITH LIMITED DATASET

This is a continuation of the experiment from Section 5.2.1. Table 7 contains evaluations of the baseline, the alpha-renaming model, and the proposed model trained with a severely limited number of samples: 80,000 instead of 799,909. We kept the number of epochs constant, and as a result, the number of training steps were also divided by ten (approximately).

The result of limiting the number of training samples is similar to the dataset perturbation, albeit much less pronounced for the baseline model. Instead of seeing the performance of the baseline model plummet as in the perturbation experiment, we observe that all models trained on the limited dataset perform similarly in terms of correctness ratio. The biggest difference is observed in the alpha-covariance values, particularly in the 5 AP category, whose ranking aligns with the perturbation experiment.

Table 7: Evaluation of the baselines and our method trained on different versions of LTLRandom35. The same results from Table 2 are shown for easier comparison. The alpha-renaming baseline was trained using 5 AP embeddings since vocabulary generalization is not valuated here. First two columns denote the training dataset and the model. Next two columns indicate the ratio of the correct predictions and exact matches on 99,989 test set samples as evaluated by `spot`. Last three columns display mean alpha-covariance values for varying atomic proposition (AP) counts, evaluated on all alpha-equivalent variants of 1000 test samples.

| Training Dataset | Model | Evaluation | | Alpha-Covariance | | |
|---|---|---|---|---|---|---|
| | | Correct | Exact | 3 AP | 4 AP | 5 AP |
| Normal | Baseline | 98.23% | 83.23% | 96.87% | 95.86% | 91.80% |
| Perturbed | Baseline | 34.13% | 12.12% | 64.93% | 57.99% | 40.91% |
| Perturbed | Alpha-Renaming | **97.96%** | **77.66%** | **99.55%** | **99.49%** | **98.86%** |
| Perturbed | Proposed | 95.94% | 76.45% | 97.66% | 97.76% | 98.29% |
| Limited | Baseline | 87.47% | 63.61% | 94.37% | 91.70% | 85.64% |
| Limited | Alpha-Renaming | **89.50%** | **64.15%** | **99.02%** | **98.67%** | **97.82%** |
| Limited | Proposed | 87.32% | 59.04% | 97.94% | 96.12% | 94.34% |

