# OpenReview forum: "Interchangeable Token Embeddings for Extendable Vocabulary and Alpha-Equivalence"
_ICLR.cc/2025/Conference — Submitted to ICLR 2025_

### Official Review · Reviewer_qigp · 2024-10-31

**Soundness:** 1
**Presentation:** 1
**Contribution:** 1
**Rating:** 3
**Confidence:** 3

**Summary:**

The subject of the paper is studying a method for using transformers trained to solve a task T on vocabulary V in order to solve the same task on a vocabulary V’ different from V (possibly of larger cardinality).

**Strengths:**

The main strength of the paper is to put the eye on a problem scarcely studied, namely how could language models be effectively used in vocabularies different from the one they were trained with.

**Weaknesses:**

The main argument put forward by the authors is that this is needed for dealing with linear temporal logic (LTL) satisfiability where a transformer is trained to output traces that satisfy the formula given as input (the empty trace meaning the formula is not satisfiable). The argument here is that unknown propositional symbols out of vocabulary V used to trained the transformer can be used in formulas at prediction time. The other “abstract” application mentioned is copying with an extendable vocabulary. However, this problem is not related to any concrete one that actually requires handling an a priori unknown vocabulary.

The rationale behind the first application is that propositional symbols not in V could actually be replaced by  others in V without changing satisfiability. This is called interchangeability in the paper. However, this concept is not formally defined but loosely presented in the introduction. Moreover, the paper states that is the same as alpha-conversion. This statement is at least cumbersome since alpha-conversion applies in the case of bound variables, but there are no quantifiers in LTL, nor in the extendable vocabulary case. For LTL, it seems to be renaming under some constraints, but not alpha-conversion. For the second application, it is not even clear where renaming occurs.

The introduction suggests that the vocabulary V' used at evaluation time is unknown at training time. However, the experiments make clear that the method requires fixing the size of the expected vocabulary at training time as well as the translation from input symbols to tokens. This is never explicitly said in the paper.

Overall:
- The paper lacks a precise definition and formalization of the problem it intends to solve.
- The experiments are not sufficiently well presented and analyzed.

**Questions:**

- Formally state the problem in the paper and provide useful examples to illustrate the problem and the intended use cases.

In Sec. 4. the authors refer to the addressed problem as “learning semantically equivalent but distinguishable (alpha-equivalent) token”. However, this problem is actually not precisely defined elsewhere. As we said before, the usual concept of alpha-conversion does not seem to apply for the intended applications foreseen in the paper.

- Justify and explain the need for partitioning the set of tokens.

In Sec. 4.1. the paper presents an embedding matrix where tokens are partitioned into so-called “regular” or “not-interchangeable”, and “interchangeable” ones. The notion of regular token is only used in this section. For the experiments in Sec. 5.1. all tokens are considered interchangeable. No mention at all of any partition of tokens in the case of LTL, though it seems there is none.

- Clarify whether the embedding has the size of the largest expected vocabulary.

The experiments in Sec. 5. (summarized in Table 3) talk about d_beta which is the number of columns of the embedding for the randomly generated part, but there is no specification of the numbers n and m corresponding to the number of regular and interchangeable tokens. The description of the experiments talk about “unique characters” but it is not clear whether they are randomly generated or fixed, or if the training and evaluation sets of characters intersect or not.

The text in L. 275 “next-token predicted probabilities … are expected to align with the input string exactly” is misleading. What do you mean by “align”. In the description of the experiments in Sec. 5.1. it looks like the number of rows of the embedding is the size of the largest expected vocabulary V’, while the transformer is only trained on symbols from V. This is not clearly stated before.

- Justify the evaluation metric in Sec. 5.1.

The evaluation metric “edit distance” is not justified. Since the validation strings are randomly generated and all tokens are interchangeable, while a token different from the one in the validation string would be considered an edit error? Also, I found Figure 3 (as well as Fig. 1 in the introduction) difficult to understand.

- Clarify questions on the perturbed dataset and baseline model of Sec. 5.2.

The experiments on LTL satisfiability are not clear (Sec. 5.2.1, Table 4).

First, the way the perturbation of the dataset is carried out is very roughly described. Do you rename both input formulas and output traces? Renaming cannot be just random since you have to preserve satisfiability and the validity of the output. Please explain.

Also, I don’t understand why fixing the order of the first occurrence is useful. Please clarify .

The text suggests it is not a model trained by the authors for the purpose of this paper. So, first of all: where does it come from? is it a model publicly available? Also, how is it capable of dealing with propositions which are not in the original set of AP? Or, are you considering the same set of AP?

The experiments in 5.2.1 and 5.2.3. do not seem to partition the tokens into regular and interchangeable. Please clarify.

- Clarify questions about alpha-covariance.

Sec. 5.2.2. Pg. 9. L 445-450 seems to suggest that there is some relation between the alpha-covariance metric and semantic correctness. It is not clear to me that the experiments in Table 4 evidence that. Please clarify.

Other questions and comments

Pg. 1. Starting Section 1 with a figure is distracting. Also, it is not related at all with the introduction and hard to understand. I think those figures do not help to clarify what is going on.

Pg. 1. The introduction gives the impression that there is no other path to solving the stated problem of satisfiability of LTL modulo alpha-conversion with Transformers than the one proposed by the paper. For instance, it could be possible to just apply a renaming to the Transformer’s alphabet before-hand. Please comment on this.

Pg. 2. endocer -> encoder

Pg. 4. Line 167. Why is it trivial to support multiple sets of different interchangeable tokens and what would be the need for such kind of embedding? Please clarify.

Pg. 5. Line 254. What do you mean by log smearing?

Pg. 5. Line 255. “clamp the scale value..”. I think bound/clip would be preferable.

Pg. 7. Line 324. Why f_bn, f_fn and AdaCos are boolean properties?

Pg. 8. Line 396. I think “empirical evidence” is more appropriate than “empirical proof”.

---

> ### Author Response · Authors · 2024-11-22
>
> We thank the reviewer for their thorough analysis. We understand if our paper was difficult to follow due to the lack of a formal problem definition. We have addressed this in the revised version.
>
> We admit that our usage of the term “alpha-conversion” is somewhat unconventional in the LTL context since there are no bound variables. However, we preferred this term because the underlying principle (renaming variables does not alter meaning) is essentially the same. Although there are no bound variables in LTL, for a given formula-trace pair, the atomic propositions (APs) can be considered “bound” to the pair. Because renaming an AP in both the formula and the trace (provided that there is no name collision) retains the same relationship between them.
>
> > ...the experiments make clear that the method requires fixing the size of the expected vocabulary at training time as well as the translation from input symbols to tokens.
>
> We now briefly clarify this topic in the newly added problem definition text. In principle, the proposed approach does not impose any restrictions about the size of the expected vocabulary. However, in practice, the maximum vocabulary size changes as a function of the number of embedding dimensions. Therefore, the expected size of the largest vocabulary needs to be considered (coarsely) while setting the hyperparameters.
>
> > The experiments in Sec. 5. (summarized in Table 3) ..., but there is no specification of the numbers n and m corresponding to the number of regular and interchangeable tokens.
>
> Since all tokens are interchangeable in the copy task, the number of unique characters is equal to the number of interchangeable tokens. The text specifies the number of unique characters for these experiments.
>
> > The description of the experiments talk about “unique characters” but it is not clear whether they are randomly generated or fixed, or if the training and evaluation sets of characters intersect or not.
>
> For each dataset, the set of unique characters is fixed. The strings are randomly generated from this set. We could not understand the comment regarding randomly generating the set of unique characters. We will be happy to clarify, if needed, if the reviewer can elaborate their concern.
>
> The training vocabulary is a subset of the evaluation vocabulary, as explained in the new problem definition. However, the particular characters are not important for the proposed method, e.g., the set of interchangeable tokens could be changed from {a, b, c} to {d, e, f} but this wouldn’t necessarily affect the embeddings (apart from the effects of randomness).
>
> > The text in L. 275 “next-token predicted probabilities … are expected to align with the input string exactly” is misleading. What do you mean by “align”.
>
> In the model’s predicted probability distribution for the token at the $i$th position, the character at the $i$th position in the input sequence must have the highest probability. We have rewritten this sentence as follows for clarity: “A string is given as input, and the model is expected to produce the input string exactly via autoregressive generation.” Since we use greedy sampling in this section, these statements are equivalent.
>
> > The evaluation metric “edit distance” is not justified.
>
> Two tokens are considered interchangeable if swapping them with each other in both the input and the output results in a semantically equivalent input-output pair. For example, suppose that the input string is “abc” in the copying task. The output is also expected to be “abc”. Since all tokens are interchangeable in this task, the input-output pair (“abc”, “abc”) is considered to be “alpha-equivalent” to (“def”, “def”), or any other pair that contains a string with three unique letters. (Note that none of the letters have any special meaning for the purpose of copying; they just need to be replicated as-is.) However, if the model outputs “def” in response to “abc”, this is considered incorrect, because both the input and the output must be transformed in the same way for alpha-equivalence. We hope that this explanation clarifies the justification of using the edit distance as an evaluation metric.
>
> > First, the way the perturbation of the dataset is carried out is very roughly described. Do you rename both input formulas and output traces? Renaming cannot be just random since you have to preserve satisfiability and the validity of the output.
>
> Yes, in each input-output pair, APs in both the formula and the trace are renamed in the same way. This applies to all “alpha-conversions” in the paper. This ensures that the transformed pair is still a valid sample, even if the renaming is random. Note that the names of the APs are arbitrary. As long as the renaming does not introduce any name collisions, it cannot break the validity of the sample.

---

> ### Author Response · Authors · 2024-11-22
>
> > I don’t understand why fixing the order of the first occurrence is useful.
>
> Fixing the order of the first occurrence introduces a perturbation into the dataset. After training the models on this dataset, we evaluate them on unperturbed samples, which are now out-of-distribution for the models. Since the baseline model overfits to the idiosyncrasies of the perturbed training dataset, its performance plummets on unperturbed samples. However, due to the construction of our embeddings, our model maintains the same performance as if it were trained on unperturbed samples, thereby demonstrating that our method introduces a robust inductive bias.
>
> > The text suggests it is not a model trained by the authors for the purpose of this paper. ... is it a model publicly available? Also, how is it capable of dealing with propositions which are not in the original set of AP?
>
> We trained all baseline models from scratch for the purposes of this paper. To explain this, we added a paragraph after the first paragraph in the “LTL Solving” subsection.
>
> The set of APs in the alpha-covariance experiment remains the same in training and evaluation because vocabulary generalization is not the focus of that experiment.
>
> > The experiments in 5.2.1 and 5.2.3. do not seem to partition the tokens into regular and interchangeable. Please clarify.
>
> The example in the newly added problem definition section demonstrates how the tokens are partitioned in the LTL solving task.
>
> > L 445-450 seems to suggest that there is some relation between the alpha-covariance metric and semantic correctness. It is not clear to me that the experiments in Table 4 evidence that.
>
> It was not our intention to make this suggestion. The referenced paragraph does not mention “semantic correctness”. Perhaps there is some confusion here, we will be happy to address any misunderstandings if the reviewer could explain the source of this impression.
>
> The paragraph claims that if renaming APs in the input leads to a different output (in the sense that the output is still different after undoing the renaming), then this is an unwanted behavior. Because the semantic meaning of the output should not change depending on how the APs in the input are named, as the names are arbitrary.
>
> > Starting Section 1 with a figure is distracting.
>
> We moved this figure to the experiment section where it has more relevance.
>
> > The introduction gives the impression that there is no other path to solving the stated problem of satisfiability of LTL modulo alpha-conversion with Transformers than the one proposed by the paper.
>
> We did not make this claim. Could you please specify which part of the text gave this impression?
>
> > For instance, it could be possible to just apply a renaming to the Transformer’s alphabet before-hand.
>
> We are not sure about what this suggestion means. But we added a new baseline that uses alpha-renaming as data augmentation during training, which may be what the reviewer was referring to. Otherwise, please clarify.
>
> > Why is it trivial to support multiple sets of different interchangeable tokens and what would be the need for such kind of embedding?
>
> We hope that the new problem definition section makes it easier to see why this is trivial: Instead of having a single $\mathbb{V}_i$ set, we would have multiple disjoint sets of interchangeable tokens. Each of these sets would have a different shared embedding part. Although this change is straightforward to implement, it was not our intention to claim that training such a model may not pose any unforeseen challenges. To clarify our position, we replaced the word “support” with “implement”.
>
> Here’s an example demonstrating when multiple sets of interchangeable tokens could be helpful: In a programming language, both constants and variables can be renamed, but there are subtle differences between these two concepts (e.g., the value of a constant cannot be altered). Thus, there can be two sets of interchangeable tokens: one for constants, and the other one for variables.
>
> > What do you mean by log smearing?
>
> Log semiring refers to the practice of performing operations in the logarithmic domain for better numerical stability. For example, multiplication is equivalent to addition in the logarithmic domain. However, in this part of the text, the particular application we were referring to is better known as the log-sum-exp trick, which is not strictly limited to log semiring despite being an application of it. To improve clarity, we replaced “log semiring” with “log-sum-exp trick”.
>
> > Why f_bn, f_fn and AdaCos are boolean properties?
>
> These properties denote whether the corresponding normalization or loss function is enabled. This is mentioned in the caption of the hyperparameter search table (Table 3 in the original manuscript and Table 4 in the revision).
>
> We thank the reviewer for their extensive list of suggestions regarding typos and word choices. We’ve taken these into consideration and updated the text accordingly.

---

### Official Review · Reviewer_wK8E · 2024-10-31

**Soundness:** 3
**Presentation:** 3
**Contribution:** 3
**Rating:** 6
**Confidence:** 3

**Summary:**

The paper proposes a novel embedding strategy for language models, specifically addressing the challenge of alpha-equivalence in formal languages. The authors introduce a dual-part embedding that combines a learned embedding that is shared between all alpha-equivalent tokens with a random component. The random component allows distinction between different variables without tying the variable symbol to the embedding. This creates an embedding technique that is agnostic to the naming of variables and enables the model to generalize to more (alpha-equivalent) variables easily. The authors evaluate on a toy task (trace copying) to empirically show the effectiveness of their approach. A second evaluation is performed on the trace generation task (from LTL specifications) to show its applicability in more complex tasks. Their results show that such an embedding indeed makes the model robust to renaming and shows the ability to handle more atomic propositions without additional training data.

**Strengths:**

The dual-part embedding approach is thoughtfully designed. It enforces shared semantics with a learnable component while introducing a random vector for differentiation. This elegantly handles alpha-equivalence, enabling the model to generalize across renamed and unseen variables.

The relevance of their work is understated, as this approach can potentially improve many neuro-symbolic systems. Alpha-equivalence/bounded variables are a core concept of formal languages. This approach can easily transfer to all neuro-symbolic tasks that combine formal languages and language models.

The paper’s structure is clear; the reader is well-guided through the method and experiments. The experiments are thorough and include an extensive ablation study. The trace copying and generation tasks clearly show the effectiveness of this approach.

**Weaknesses:**

While the experiments are extensive, it would help if the paper clarified the specific contributions of each experiment to the overall findings. The following is more detailed and specific feedback on the individual experiments.

__Baseline__. All experiments (Table 4, Figure 1, and Figure 4) mention a baseline approach. The authors should clarify how this baseline is created and how it relates to related work (Hahn et al.). I assume the authors replicated experiments from Hahn et al., but it would be beneficial if the authors would clarify where and how the replication's performance compares to the related work.

__Training Steps__. The authors do not provide the training time, hardware, number of epochs/steps, etc., for any of the experiments and baselines. Comparing the training epochs/steps of their approach to the baseline would be especially important to assess the effectiveness of their approach.

__Hyperparameter Search__.The ablation study on hyperparameter choices (5.1.2) shows thoroughness and dedication. However, the authors themselves note that the results (Table 4) are inconclusive on which hyperparameter choice is the best because of the high variance. They then extensively cross-test the correlation coefficient to see which hyperparameter choices have a positive effect. Contrary to the findings of these extensive experiments, the authors fall back to Table 4 to declare their best model (which partly contradicts the correlation coefficients). Despite the extensive experiments, this feels inconclusive and inconsistent. To improve clarity, it might be beneficial to move the ablation details to the appendix and present the “best” model configuration in the main text, leaving more space to highlight the generalization to additional APs on the trace copying task.

__Immunity to Renaming__. The phrase “empirical proof” (line 398) is misleading since empirical experiments provide evidence rather than proof. Additionally, if understood correctly, the embedding of a variable/token does not depend on the actual variable symbol/token. Therefore, for example, the model cannot distinguish whether a variable is named a or b, only that they are different. This means that the model is - by construction - immune to variable renaming. If the model is - by construction - immune to variable renaming (during training and evaluation), there is no necessity to empirically validate this property. On the contrary, the Alpha-Covariance results on the “Limited” experiment (proposed model) seem to invalidate this property as the Alpha-Covariance seems to degrade with more AP. Can the authors comment on this observation?

Instead, focusing on other non-obvious generalization aspects would further strengthen the experimental section:

__AP Generalization__. The most important experiment is the generalization to more APs (5.1.2 and 5.2.3). The paper's main contribution is showing that the proposed embedding improves generalization to more APs, especially on a non-trivial task (5.2.3). However, the comparison with the baseline model might be biased. While it does show that the proposed model generalizes, they don’t fully establish that the baseline model can’t generalize in this way. The experiment instead shows that Transformers cannot generalize to unseen tokens, which is a well-known fact. A more balanced comparison could involve ensuring that all 30 (or 10) APs appear in the baseline’s training set. Note that this does not imply that the dataset would contain samples exceeding 5 APs, only that all tokens are seen during training. Since the authors admit that the baseline model performs better on in-distribution evaluation, the outcome of such an experiment is open.

__Minor Details__:
 - Adding axis legends for edit distance in the heatmaps (Figures 1, 3, 4) and labeling subfigures (e.g., a), b), c)) would improve understanding and make the experiments easier to reference. Placing Figure 1 in the evaluation section, where it has more context, might also enhance the flow as the Figure is hard to understand without the necessary context of the evaluation section.
 - The grey area in Figure 3 represents impossible data points, and the green box (boundaries of the training dataset) overlaps this region. If these data points cannot exist, perhaps the green box boundaries could be adjusted to prevent visual confusion.

__Summary__. Overall, this paper introduces a well-motivated and potentially impactful approach, and it’s clear that substantial engineering effort went into the experiments. However, the experiments do not sufficiently showcase the advantage over the baseline. This does not imply that the results are not good enough but rather that comparisons to the baselines need to be chosen more carefully. Furthermore, a second application from a different formal language domain than LTL would strongly improve the paper’s relevance in the area.

**Questions:**

Please address the comments above.

---

> ### Author Response · Authors · 2024-11-22
>
> We thank the reviewer for their elaborate comments.
>
> > The authors should clarify how this baseline is created and how it relates to related work (Hahn et al.). I assume the authors replicated experiments from Hahn et al., but it would be beneficial if the authors would clarify where and how the replication's performance compares to the related work.
>
> We have trained several models to replicate the results from Hahn et al. (2021). We used the best hyperparameters they report, but we also experimented with the following changes:
> Using RoPE for positional encoding instead of sinusoidal positional encodings in the decoder. This increased the correct rate from 97.8% to 98.0% on the validation set.
> Using the AdaCos loss function and feature normalization $f_{fn}$. The correct rate on the validation set increased to 98.2% when combined with RoPE.
> After determining the best baseline model on the validation set, we evaluated it on the test split of LTLRandom35 and obtained a 98.2% correct rate against the 98.5% reported by Hahn et al. (2021).
>
> To explain this in the paper, we added a paragraph after the first paragraph in the “LTL Solving” subsection (page 8, line 378).
>
> > The authors do not provide the training time, hardware, number of epochs/steps, etc., for any of the experiments and baselines.
>
> We provided the batch size and training steps in Table 3 (Table 2 in the original manuscript) for all experiments. The training steps are the same for all models within the same experiment. The hardware used in the experiments was varied, and as a result, the training times are not directly comparable. However, we used the same hardware (single NVIDIA H100) for the models that were trained on the perturbed dataset (Table 2). The baseline model took 2 hours 10 minutes, and our model took 2 hours 11 minutes. This demonstrates that the overhead of our method is negligible.
>
> **Hyperparameter search.** We agree that our hyperparameter search exhibits high variance despite our efforts. As the reviewer suggested, we moved this section to Appendix and only presented the best models in the main text.
>
> **Immunity to renaming.** In accordance with the reviewer’s suggestion, we renamed “empirical proof” to “empirical evidence”. As they mentioned, our method can be considered “immune to renaming” by construction in the sense that changing a variable name does not necessarily change its embedding. However, due to randomness, different variables—and perhaps even the same variable across different runs—will have different embeddings.
>
> In the alpha-covariance experiment, we generate a random embedding for each AP token once at the start, and then observe the output of the model for each renaming permutation. Thus, alpha-conversions in this experiment are equivalent to shuffling the random embeddings in our method. As a result, the alpha-covariance experiment measures our model’s robustness against differences in random embeddings. We have added this explanation to the revised text (the second last paragraph in Alpha-Covariance section, lines 448-452).
>
> **AP Generalization Baseline.** We have added the baseline that the reviewer recommended to the generalization experiments (Sections 5.1.2, 5.1.4, 5.2.3). We observe that our method outperforms this baseline on the LTL solving task. We added a new paragraph (on page 10) to discuss the results.
>
> > Adding axis legends for edit distance in the heatmaps (Figures 1, 3, 4) and labeling subfigures (e.g., a), b), c)) would improve understanding and make the experiments easier to reference.
>
> In each of these figures, all heatmaps share the same y-axis. Consequently, in the original manuscript, we showed the y-axis legend only for the leftmost heatmap. This allows us to place the heatmaps closer to each other for easier comparison. In the revision, we have either added the axis legends or clarified that the y-axis is shared in the caption for all heatmap figures.
>
> We appreciate the suggestion, but we did consider adding subfigure labels upon the suggestion but ended up  avoiding them for simplicity, as each heatmap already has a title.
>
> > Placing Figure 1 in the evaluation section, where it has more context, might also enhance the flow as the Figure is hard to understand without the necessary context of the evaluation section.
>
> We have applied this change in the revision.
>
> > The grey area in Figure 3 represents impossible data points, and the green box (boundaries of the training dataset) overlaps this region. If these data points cannot exist, perhaps the green box boundaries could be adjusted to prevent visual confusion.
>
> The y-axis of the green box represents the number of unique characters in the dataset. Therefore, although it overlaps the impossible region, it represents that the dataset features 30 unique characters across different samples. We have updated the explanation in the figure’s caption to clarify.

---

> > ### Comment · Reviewer_wK8E · 2024-11-27
> >
> > With the hyperparameter search in the Appendix, the paper now focuses more clearly on the main experiments and is easier to follow.
> >
> > The additional experiment on alpha-renaming the baseline confirmed two things:
> >  - The original baseline was indeed not a fair comparison, as it shows that the bad performance of the original baseline is attributed to unseen tokens.
> >  - The proposed model is still significantly better than the alpha-renaming baseline, showing the true advantage of the proposed method over naive implementations.
> >
> > This experiment notably strengthens the paper.

---

> > > ### Author Response · Authors · 2024-11-27
> > >
> > > We sincerely thank the reviewer for their thoughtful engagement and valuable feedback throughout the review process. We are pleased to hear that the manuscript is now stronger.
> > >
> > > While we believe that the discussion has addressed the raised concerns, we would be happy to answer any further questions. Thank you again for your constructive and encouraging feedback.

---

> > > ### Author Response · Authors · 2024-11-28
> > > **The Second Revision**
> > >
> > > Since you emphasized the importance of the alpha-renaming baseline, we added it to the dataset perturbation experiment in the second revision. Moreover, we evaluated alpha-covariance in the LTL generalization experiment to complement the aforementioned experiment. Thank you for your attention and time.

---

### Official Review · Reviewer_xSvQ · 2024-11-01

**Soundness:** 2
**Presentation:** 3
**Contribution:** 2
**Rating:** 3
**Confidence:** 5

**Summary:**

Most formal languages, including the lambda calculus, include bound variables, e.g. the variable $X$ in $\forall$ X. human($X$) $\implies$  mammal($X$).  Terms in such languages are held to be equivalent under renaming of bound variables, a principle known as alpha-equivalence.  E.g. we can rename $X$ to $Y$ in the above expression without changing its meaning.  However, even though the actual name does not matter, an LLM which does logical reasoning must still be able to distinguish between different names, e.g. the variables $X$ and $Y$ refer to distinct entities in $\forall X,Y$. parent_of($X$, $Y$) $\implies$ child_of($Y$, $X$).

The authors propose a novel embedding scheme for handling variables that are subject to alpha-renaming.  For each such variable, one part of the embedding is a normal learned embedding meaning "variable", while the second part is randomly generated for each training example, but is consistent across all uses of that variable within the example.  I.e. the embedding of $X$ is partially random, but all occurrences of $X$ within the training example use the same embedding for $X$.

The authors test their scheme on a synthetic copying task, and on a synthetic LTL (temporal logic) task involving randomly generated formulae.

**Strengths:**

The issue of alpha-equivalence, and of extensible vocabularies in general, has been underexplored in the literature, and the author's proposed solution seems principled and well-designed.  The authors describe their technique clearly, and cite the relevant literature.

Experimentally, the most interesting results of the paper by far are the generalization results.  The authors proposed solution generalizes to longer sequences and longer vocabulary in the synthetic copying task, and to longer formulae using more variables for the LTL task, whereas the baselines do not.  IMO, these generalization results are important.

**Weaknesses:**

Unfortunately, most of the authors experiments were not very convincing to me.  The copying task is essentially a unit test; it merely shows that the embeddings are working as intended.  The LTL perturbation task deliberately trains the baseline model in such a way that it is forced to operate out-of-distribution, so there's no surprise that the baseline does poorly.
Some of the other tests, such as different randomization schemes, seem uninteresting and/or irrelevant to me.

Most importantly, both of the main tasks were entirely synthetic, which means that the variable names had no actual meaning.  In real-world human-written code or logical proofs, names typically do have meaning.  Meaningful names are a double-edged sword.  On the one hand, they may prevent the model from recognizing certain equivalences.  On the other hand, they can also guide the model in reasoning about the proof or code; a variable named "grocery_list", for example, encodes a lot of information about how that variable is likely to be used.  Names are certainly crucial for human reasoning; typical code-obfuscation techniques work by renaming all the variables, which prevents humans from understanding the code.

To really demonstrate that the author's technique is effective in practice, I would want to see it evaluated on something other than a toy synthetic dataset, such as a code-repair task for actual human-written code.  Unfortunately, the authors do not even bother writing a discussion section about the pros & cons of erasing names from LLM input; the paper would be stronger if it at least had that.

In the short term, a way to improve the paper would be to compare the technique against other alternatives.  For example, DeBruijn indices are another common way of encoding formulae, which eliminate the need for alpha-renaming.

Another potential comparison would be to use a conventional subword vocabulary for variable names, but to train the baseline model on formulae that have already undergone random alpha-renaming of variables, so that the baseline model is robust against renaming -- that would be a more interesting result than the contrived perturbation experiment in this paper.

Although it would require training a larger model, a good way to show the importance of alpha-equivalence in practice would be to train a conventional model on human-written proofs or code, and then show that the conventional model can be forced to make unsound logical deductions if the variables are renamed.

Overall, I would say that this is promising work, but publication at this point is perhaps premature.

**Questions:**

Please see the "weaknesses" section above -- I would be particular interested in hearing the authors response to comparisons against alternative approaches such as DeBruin indices or alpha-renaming-as-part-of-training.

---

> ### Author Response · Authors · 2024-11-22
>
> We appreciate the insights provided by the reviewer.
>
> > The LTL perturbation task deliberately trains the baseline model in such a way that it is forced to operate out-of-distribution, so there's no surprise that the baseline does poorly.
>
> We agree that the LTL perturbation experiment was designed to showcase the strengths of our method. However, we believe that it’s important to empirically demonstrate that our method induces a favorable inductive bias for alpha-equivalence. Technically, the proposed model is also forced to operate on out-of-distribution samples in this experiment if we treat the model as a black-box. However, the construction of embeddings in our model allows it to process the biased samples in the same way that it processes unbiased samples.
>
> The difference between our method and traditional embeddings can be likened to the difference between convolutional neural networks (CNNs) and multi-layer perceptrons (MLPs). Anything that can be learned by a CNN can also be learned by an MLP, but CNNs present an inductive bias for translation invariance, similarly to how our method promotes alpha-covariance. Moreover, CNNs can operate on inputs larger than the training samples, resembling the vocabulary generalization capabilities of our method. To empirically verify the inductive bias of CNNs, an experimenter could remove all translation variance in the training set (similarly to the perturbed dataset in our experiment), and evaluate the models on translated samples. This is the core idea behind our LTL perturbation experiment. Clearly, however, this CNN analogy has its limits: Since our method incorporates randomness, alpha-covariance is not fully guaranteed by construction; different random embeddings can lead to different outputs. That’s why the LTL perturbation experiment is potentially more intriguing than its CNN counterpart.
>
> > I would want to see it evaluated on something other than a toy synthetic dataset.
>
> Although our first experiment task (copying with extendable vocabulary) involves a toy dataset, the second task (LTL solving) is far from being a toy example despite the fact that the dataset is synthetic. LTL solving is an important problem in the literature, and the current SOTA seems to be DeepLTL (Hahn et al, 2021), which our paper builds on. In particular, DeepLTL can solve problems on which the classical solvers time out. Hahn et al. (2021) propose another dataset (LTLPattern126) for the same task that—instead of being synthetically generated—is created from real LTL formula-trace patterns found within the literature. We would be delighted to present experimental results on this dataset as well. But unfortunately, it was not released publicly.
>
> ### Meaningful variable names
>
> The reviewer’s remarks about meaningful variable names are insightful and we agree that this will be an important consideration for future applications of our method. However, there are many applications in which meaningful variable names are not available, with one such example being the presented LTL solving problem. Even the previously mentioned LTLPattern126 dataset does not feature meaningful variable names. Since our applications do not involve meaningful variable names, we consider this to be beyond the scope of our paper. But we agree that this is an interesting direction for future work. Thus, in the revision, we have mentioned this idea in the conclusion.
>
> ### DeBruijn indices
>
> DeBruijn indices present a natural way to represent bound variables in lambda calculus without explicitly naming them. However, it is not directly applicable to the tasks we considered in the experiments section. Since DeBruijn indices depend on the structure of lambda calculus, adapting DeBruijn indices to our tasks is not straightforward. Furthermore, DeBruijn indices do not address vocabulary generalization, which was our primary motivation.
>
> ### Alpha-Renaming Baseline
>
> We have added the reviewer's suggestion as a baseline in Sections 5.1.2, 5.1.4, 5.2.3 (also see the “Baselines” paragraph on page 5, line 250). We have also added a paragraph to discuss the results (page 10, line 486).
>
> ### Human-written proofs or code
>
> We thank the reviewer for this suggestion. While training a model on human-written proofs or code to demonstrate the impact of alpha-equivalence is clearly an interesting idea, it is beyond the scope of our current work. Our focus was on tasks where human-written variable names are unavailable, such as the LTL solving task. Training a model on human-written data would require not only a shift in focus but also substantially more computational resources due to the scale and complexity of such datasets. We appreciate the reviewer highlighting this direction and agree that it represents an intriguing area for future research.
>
> ### References
>
> Hahn, Christopher, et al. Teaching Temporal Logics to Neural Networks. arXiv:2003.04218, arXiv, 18 Feb. 2021. arXiv.org, https://doi.org/10.48550/arXiv.2003.04218.

---

> > ### Comment · Reviewer_xSvQ · 2024-11-26
> >
> > Thank you for adding the alpha-renaming data augmentation.  I definitely feel that the additional baseline strengthens the paper.  The discussion in section 5.2.3 is especially informative.
> >
> > I must admit that I'm a bit confused as to why alpha-renaming performs better than full-vocabulary on the random sequence copying task, though.  Surely random sequences generated with the full vocabulary, and random sequences generated with a restricted vocab, but in which some tokens are renamed to other tokens within the full vocabulary, are almost the same task?  The note says that the difference is that alpha-renaming only sees 5 unique tokens, but why, then, does it perform better?

---

> > > ### Author Response · Authors · 2024-11-26
> > >
> > > Thank you for your positive comments on the alpha-renaming data augmentation and for taking the time to engage with the revision. We are glad to hear that you found the additional baseline valuable and appreciated the discussion in Section 5.2.3.
> > >
> > > Regarding your question about why alpha-renaming performs better than the full-vocabulary baseline on the copying task, as mentioned in the paper (page 7, line 327), we consider the significance of this result to be questionable due to high variance across training runs. One possible explanation, however, could be that the full vocabulary baseline might have adapted to idiosyncrasies of the training dataset, which could have arisen due to the pseudo-random generator. The alpha-renaming baseline, on the other hand, is more resilient to such peculiarities thanks to data augmentation.
> > >
> > > Thank you again for your thoughtful review and for highlighting this aspect of our work.

---

> > > ### Author Response · Authors · 2024-11-28
> > > **The Second Revision**
> > >
> > > To investigate the alpha-renaming baseline further, we added it to the dataset perturbation experiment and evaluated the alpha-covariance performance of the LTL generalization models in the second revision. Since LTL solving is a real task with less variance across training runs, these new experimental results may lead to a better understanding of how the alpha-renaming baseline differs from the full vocabulary baseline. Thank you for your attention and time.

---

### Official Review · Reviewer_BxyR · 2024-11-03

**Soundness:** 2
**Presentation:** 1
**Contribution:** 2
**Rating:** 3
**Confidence:** 3

**Summary:**

The paper presents an approach to create embeddings for LTL (Linear-time Temporal Logic). The main challenge that is addressed is to represent tokens during inference that are not seen during training. The idea is to represent interchangeable tokens during inference instead of learning the representations during training. A part of the embedding is used to represent the similarities and tie tokens together and a second randomized part distinguishes between interchangeable tokens. Experiments are performed with large vocabularies showing that the proposed approach can learn representations for interchangeable tokens more accurately.

**Strengths:**

- Generalizing representations to new tokens to represent the alpha equivalence seems like a relevant and significant problem in the domain of LTL
- The method seems to be intuitive and scalable as shown by the experiments

**Weaknesses:**

- Most of the contribution is based on empirical results, since the novelty of the algorithmic contribution itself is perhaps not as strong (e.g. generating the random embeddings). Given this, it felt like the experiments were not particularly strong and also presented in a way that is a little hard to follow. Specifically, the first experiment is synthetically designed with random strings. Further, it is not compared to any other method but only on variants of the same approach, so it is hard to judge whether the results are significant or not. The second dataset seems like an existing benchmark and it is not clear what the “baseline” is in the case. Also, there is a perturbed and limited dataset which again is not very clear as to what is the difference is between the two. Regarding the metric, it seems to introduce a new metric, but there is very little context to justify the metric, for e.g. are there other types of metrics, what are the trade-offs etc. In general, there should a bit more discussion on this.
- Regarding the method itself, the randomization may introduce a lot of variance, so how would this variance be accounted for in the results presented?
- I was not sure what N.D., H.V. ,etc. were showing in one of the tables
- The presentation could be improved with some examples and real-world motivation of LTL and relating them with the empirical results

**Questions:**

- How significant are the empirical results (please see under weaknesses for details)
- Are there any real-world cases that could be evaluated for showing the value of the alpha-equivalence and the significance of the proposed methods as well as limitations or trade-offs.

---

> ### Author Response · Authors · 2024-11-22
>
> We thank the reviewer for their feedback.
>
> > Most of the contribution is based on empirical results, since the novelty of the algorithmic contribution itself is perhaps not as strong (e.g. generating the random embeddings).
>
> To the best of our knowledge, no other work in the literature attempts to create an extensible embedding for interchangeable tokens, and our dual-part embedding method is novel, as far as we are aware. If the reviewer is aware of any prior work that aligns with this approach or presents similar contributions, we would be grateful if they could provide references to help us better position our work within the existing literature.
>
> Although generating random embeddings is a popular technique for initialization, it constitutes only a part of our method and is used for a different purpose. Furthermore, generating the random part in our embeddings entails special considerations that led us to propose several random generation methods.
>
> > Specifically, the first experiment is synthetically designed with random strings. Further, it is not compared to any other method but only on variants of the same approach, so it is hard to judge whether the results are significant or not.
>
> Copying task is an important synthetic problem for autoregressive models because many real-life tasks involve copying as a subtask, e.g., when asked to fix a bug in a given code snippet, an LLM must copy most of the input as-is, and only alter the faulty part. The transformer models are better than many other models in this task (Jelassi et al., 2024) since the attention mechanism has access to the full context. As a result, we only considered transformer-based models as baselines.
>
> > The second dataset seems like an existing benchmark and it is not clear what the “baseline” is in the case.
>
> We have added a paragraph in Section 5 and another in Section 5.2 to explain the baselines. This dataset is not a benchmark. We apply our embeddings to DeepLTL (Hahn et al., 2021), which approaches the LTL solving problem using a language modeling perspective.
>
> > Also, there is a perturbed and limited dataset which again is not very clear as to what is the difference is between the two.
>
> We kindly refer to the following parts in the first submitted manuscript:
> 1. Lines 395-397 (lines 387-392 in the revision): “To demonstrate that our method creates a helpful inductive bias, we created a perturbed version of the LTLRandom35 dataset. In particular, we rename the APs such that the order of the first AP appearances in the trace is always the same.”
> 2. Lines 421-422 (lines 415-418 in the revision): “Table 4 (Table 2 in the revision) also contains evaluations of the baseline and the new model trained with a severely limited number of samples: 80,000 instead of 799,909.”
>
> We would be more than happy to explain further if the reviewer can please specify the unclear aspects.
>
> > Regarding the metric, it seems to introduce a new metric, but there is very little context to justify the metric, for e.g. are there other types of metrics, what are the trade-offs etc. In general, there should a bit more discussion on this.
>
> We are not aware of any other metric to measure a model’s robustness to alpha-conversions. The proposed metric is straightforward: It simply counts the variation in the outputs of the model corresponding to the alpha-equivalent pairs and normalizes this value to [0, 1] range (since the total number of alpha-equivalent pairs depends on AP count), with 1 being the most desirable value.
>
> In the revision, we have updated the “Alpha-Covariance” section to emphasize that there is no other metric available (to the best of our knowledge) and provide more justification (e.g., line 433: “The model's sensitivity to alpha-conversions could be quantified by simply $|\mathbb{U}|$, but this value may be hard to interpret since it depends on $|\mathbb{P}|$.”). Please keep in mind that both the original submission (page 9, lines 445-450) and the revision (page 9, lines 441-446) features an intuitive explanation about the metric.
>
> ### References
>
> Hahn, Christopher, et al. Teaching Temporal Logics to Neural Networks. arXiv:2003.04218, arXiv, 18 Feb. 2021. arXiv.org, https://doi.org/10.48550/arXiv.2003.04218.
>
> Jelassi, Samy, et al. Repeat After Me: Transformers Are Better than State Space Models at Copying. arXiv:2402.01032, arXiv, 3 June 2024. arXiv.org, https://doi.org/10.48550/arXiv.2402.01032.

---

> ### Author Response · Authors · 2024-11-22
>
> > Regarding the method itself, the randomization may introduce a lot of variance, so how would this variance be accounted for in the results presented?
>
> 1. In Section 5.1, we repeat the evaluation 10 times and report the average, as explained in the “Evaluation method” paragraph.
> 2. Section 5.1.3 “Sensitivity to randomness in embeddings” is dedicated to investigating this question.
> 3. In the alpha-covariance experiment (Section 5.2.2), we generate the embeddings once at the start for our method. This means that the alpha-conversions in this experiment are equivalent to shuffling the random embeddings in our method. Therefore, the alpha-covariance measures our model’s robustness against differences in random embeddings.
>
> > I was not sure what N.D., H.V. ,etc. were showing in one of the tables
>
> The table that the reviewer is referring to is an in-text table at the top of page 7 in the original manuscript. The explanation for these acronyms can be found in line 330 in the original paper, immediately after the table. However, this table is now in the appendix after the revisions.
>
> > The presentation could be improved with some examples and real-world motivation of LTL and relating them with the empirical results
>
> The newly added problem definition in the revised manuscript now features examples.
>
> > How significant are the empirical results (please see under weaknesses for details)
>
> We think that our experiments successfully demonstrate the following:
> 1. Our embeddings can generalize to larger vocabulary sizes. This is seen in Sections 5.1.1, 5.1.2, 5.1.4, and 5.2.3. The significance of the result is more pronounced in larger experiments, such as the bigger copy task and the LTL solving task.
> 2. Our embeddings create an inductive bias for learning alpha-equivalence (Sections 5.2.1 and 5.2.2). Specifically, our model maintains a good performance despite the perturbation while the performance of the baseline model plummets under perturbation. This indicates that our method introduced a robust inductive bias.
>
> > Are there any real-world cases that could be evaluated for showing the value of the alpha-equivalence and the significance of the proposed methods as well as limitations or trade-offs.
>
> We would like to highlight that the LTL solving application is a real, important problem in the formal verification literature. DeepLTL (Hahn et al., 2021), which is the method that we build on in the LTL solving subsection, is able to solve formulae on which the classical solvers time out.
>
> The value of alpha-equivalence can be intuitively understood in formal languages such as LTL: Since renaming the APs does not alter the meaning, changing the names of the atomic propositions (APs) in the input should not affect the model’s output in an ideal world.
>
> The concept of alpha-equivalence can be likened to translation in the vision domain. Translation invariance is desired in applications where the location of the object within the image does not matter. Likewise, alpha-covariance is desired when interchangeable tokens exist in a language modeling problem. Similarly to how convolutional neural networks (CNNs) introduce an inductive bias for translation invariance, our method facilitates learning alpha-covariance.
>
> Regarding the tradeoffs, in the conclusion section, we highlighted how the performance on in-distribution samples decreases slightly in our method (line 510 in the original manuscript, line 529 in revision), as the results in Table 2 indicate. We think this is an example of the well-known bias-variance tradeoff.

---

> > ### Comment · Reviewer_BxyR · 2024-11-27
> >
> > Thanks a lot for the detailed feedback and the revised version which definitely improved the clarity of the work and my own understanding of the work. I did have a question regarding the LTL task which as you and others indicate seems to be the core task that the proposed approach has made contributions to advance. I studied the original paper you cited by Hahn et al. and they seem to show their approach on 6 different datasets (with different training and testing combinations). My question is is there a reason why only of these (LTLRandom35) was chosen to show that the proposed approach improves robustness to perturbations while the original LTL does not. What is the limiting factor for the other datasets? Showing these on a more broader selection (if indeed this was possible) could have greatly strengthened the empirical claims regarding generality of the inductive bias.
> >
> > Thanks

---

> > > ### Author Response · Authors · 2024-11-27
> > >
> > > Thank you for taking the time to read the revised version and our response. We are glad to hear that the revision and our comments were helpful.
> > >
> > > > I studied the original paper you cited by Hahn et al. and they seem to show their approach on 6 different datasets (with different training and testing combinations), (...) is there a reason why only of these (LTLRandom35) was chosen?
> > >
> > > We thank the reviewer for highlighting this point.
> > >
> > > First of all, we would like to emphasize that our goal is not to improve DeepLTL (Hahn et al., 2021) but to develop and demonstrate vocabulary generalization capabilities and inductive bias for alpha-equivalence. Clearly, however, LTL solving is one of the many interesting applications to evaluate the proposed approach.
> > >
> > > We are assuming that the reviewer is referring to these training and test set combinations from Hahn et al. (2021):
> > > 1. LTLRandom35, LTLRandom35
> > > 2. LTLRandom35, LTLRandom50
> > > 3. LTLPattern126, LTLPattern126
> > > 4. LTLPattern126, LTLUnsolved254
> > > 5. PropRandom35, PropRandom35
> > > 6. PropRandom35, PropRandom50
> > >
> > > We use the first combination in the “Dataset Perturbations” section. Furthermore, the test dataset we created for the LTL generalization section contains longer samples with more APs. As a result, it can be considered an extended version of LTLRandom50. Specifically, the main difference between LTLRandom50 and our generalization dataset is that the AP count is limited to 5 in LTLRandom50, whereas we use 10 in our generalization experiments.
> > >
> > > We also note that among the remaining dataset pairs from Hahn et al. (2021), the datasets LTLPattern126 and LTLUnsolved253 were not released publicly. Consequently, even though we would be delighted to demonstrate experimental results on these datasets, we unfortunately did not have the opportunity to do that.
> > >
> > > The last two dataset pairs involve PropRandom35 & PropRandom50, which are used in a different task: predicting assignments for propositional logic. We did not prioritize it due to its similarity to LTLRandom. In particular, the PropRandom datasets are composed of randomly-generated synthetic samples (similarly to LTLRandom), and the task can be simply thought of as a non-temporal version of LTL solving.

---

### Author Response · Authors · 2024-11-22

We greatly appreciate the valuable feedback given by the reviewers. We have revised the paper carefully. Below we give a list of paper updates. We provide our specific answers as individual replies to the reviews.

**Updates:**

1. We moved the hyperparameter search part to the appendix as per Reviewer wK8E’s suggestion.
2. In accordance with Reviewers xSvQ and wK8E’s suggestion, we added a new baseline that uses alpha-renaming as a data augmentation strategy during training. This allows the model to learn a larger fixed embedding vocabulary on a dataset with smaller vocabulary.
   * The first heatmap on Figure 3 (proposed method on the smaller copying task trained with full length samples and limited vocabulary) was removed since it contains all zeros. The new baseline is added.
   * Figure 4 (LTL heatmaps) is updated with the new baseline.
   * The second paragraph of Section 5 (page 5, lines 250-257) explains the baseline types.
3. We added a formal problem definition section as recommended by Reviewer qigp.
4. We moved the preliminary section about language models to the appendix since it’s well-known in this field.
5. We moved Figure 1 from introduction to the experiments section, as advised by Reviewers wK8E and qigp. Also, we no longer refer to this figure in Section 1.
6. We clarified what the alpha-covariance experiment means for our method in the second last paragraph in the corresponding section. For more details, see our discussion with Reviewer wK8E.
7. After the first paragraph in the “LTL Solving” subsection, we added a paragraph to explain the LTL baselines and compare them against the literature, as a result of our discussions with Reviewers wK8E and qigp.
8. We increased the sample count in Table 2 and specified it in the caption.
9. We updated the “Alpha-Covariance” section to provide more justification for the metric in response to Reviewer BxyR’s comment. To enhance clarity, we used $|\mathbb{P}|$ instead of $n$ in Equation 2.
10. In the conclusion section, we added the Reviewer xSvQ’s suggestion (alpha-equivalence with meaningful variable names) as a direction for future work.
11. We fixed the typos and improved writing in some sections as pointed out by reviewers.

We think that these updates address the reviewers’ concerns to a great extent.

---

> ### Author Response · Authors · 2024-11-28
> **The Second Revision**
>
> Following the insightful discussion with the reviewers, we made another revision to improve the experiments and emphasize our contributions.
>
> **Updates:**
>
> 1. We added the alpha-renaming baseline to the dataset perturbation experiment (Table 2, Section 5.2.1). As noted by Reviewers xSvQ and wK8E, this new baseline constitutes a better comparison for our model.
> 2. To complement the previous point, we evaluated the alpha-covariance performance of the models in the LTL generalization section up to 10 APs (Table 3, Section 5.2.3). We commented on these results (page 10, line 508).
> 3. We moved the limited dataset experiment from Section 5.2.1 to appendix (Appendix A.4) since its results are very similar to the perturbation experiment. Furthermore, presenting these together in Section 5.2.1 has previously led to some confusion, as our discussion with Reviewer BxyR shows.
> 4. We updated the abstract and the introduction section to highlight the alpha-renaming baseline and alpha-covariance, with the intention of communicating our contributions better.
> 5. We did some minor fixes to improve writing.
>
> Overall, this update aims to improve the relevance of alpha-covariance by providing a better baseline and investigating alpha-covariance under out-of-distribution settings. We believe that these additions have significantly improved the paper, and we are grateful for the valuable suggestions and feedback provided by reviewers. We now believe that all major concerns raised in the reviews have been addressed.

---

### Meta-Review · Area_Chair_naRx · 2024-12-25

**Metareview:**

The authors propose a method for creating embeddings for linear-time temporal logics (LTL) where a challenge is that tokens that were not encountered during training may need to be used during inference. The authors propose interchangeable tokens where some part represents similarities between such interchangeable tokens and a second random part which uniquely identifies them. There is very little theory in the paper and the reviewers agree that substantial more empirical evaluation is necessary.

**Additional Comments On Reviewer Discussion:**

The reviewers engaged well with the authors and wrote very helpful reviews, including explaining some of the subtleties of formal logics.

---

### Decision · Program_Chairs · 2025-01-22

Reject